# HIV Drug Resistance Mutations Detection by Next-Generation Sequencing during Antiretroviral Therapy Interruption in China

**DOI:** 10.3390/pathogens10030264

**Published:** 2021-02-25

**Authors:** Miaomiao Li, Shujia Liang, Chao Zhou, Min Chen, Shu Liang, Chunhua Liu, Zhongbao Zuo, Lei Liu, Yi Feng, Chang Song, Hui Xing, Yuhua Ruan, Yiming Shao, Lingjie Liao

**Affiliations:** 1National Center for AIDS/STD Control and Prevention, Chinese Center for Disease Control and Prevention, Beijing 102206, China; miaomiao_ana@163.com (M.L.); zuozhongbaocdc@163.com (Z.Z.); cdcliulei@163.com (L.L.); fengyi@chinaaids.cn (Y.F.); songchang604@163.com (C.S.); xingh09@163.com (H.X.); ruanyuhua92@163.com (Y.R.); yshao@bjmu.edu.cn (Y.S.); 2Guangxi Center for Disease Control and Prevention, Nanning 530028, China; liangshujia@126.com; 3Chongqing Center for Disease Control and Prevention, Chongqing 400042, China; 2008zhch@163.com; 4Yunnan Center for Disease Control and Prevention, Kunming 650022, China; chenminyx@126.com; 5Sichuan Center for Disease Control and Prevention, Chengdu 610041, China; liangshu523@163.com; 6Henan Center for Disease Control and Prevention, Zhengzhou 450016, China; chunhua5167@163.com

**Keywords:** HIV drug resistance, sanger sequencing, next-generation sequencing, interrupted antiretroviral therapy

## Abstract

Patients with antiretroviral therapy interruption have a high risk of virological failure when re-initiating antiretroviral therapy (ART), especially those with HIV drug resistance. Next-generation sequencing may provide close scrutiny on their minority drug resistance variant. A cross-sectional study was conducted in patients with ART interruption in five regions in China in 2016. Through Sanger and next-generation sequencing in parallel, HIV drug resistance was genotyped on their plasma samples. Rates of HIV drug resistance were compared by the McNemar tests. In total, 174 patients were included in this study, with a median 12 (interquartile range (IQR), 6–24) months of ART interruption. Most (86.2%) of them had received efavirenz (EFV)/nevirapine (NVP)-based first-line therapy for a median 16 (IQR, 7–26) months before ART interruption. Sixty-one (35.1%) patients had CRF07_BC HIV-1 strains, 58 (33.3%) CRF08_BC and 35 (20.1%) CRF01_AE. Thirty-four (19.5%) of the 174 patients were detected to harbor HIV drug-resistant variants on Sanger sequencing. Thirty-six (20.7%), 37 (21.3%), 42 (24.1%), 79 (45.4%) and 139 (79.9) patients were identified to have HIV drug resistance by next-generation sequencing at 20% (v.s. Sanger, *p* = 0.317), 10% (v.s. Sanger, *p* = 0.180), 5% (v.s. Sanger, *p* = 0.011), 2% (v.s. Sanger, *p* < 0.001) and 1% (v.s. Sanger, *p* < 0.001) of detection thresholds, respectively. K65R was the most common minority mutation, of 95.1% (58/61) and 93.1% (54/58) in CRF07_BC and CRF08_BC, respectively, when compared with 5.7% (2/35) in CRF01_AE (*p* < 0.001). In 49 patients that followed-up a median 10 months later, HIV drug resistance mutations at >20% frequency such as K103N, M184VI and P225H still existed, but with decreased frequencies. The prevalence of HIV drug resistance in ART interruption was higher than 15% in the survey. Next-generation sequencing was able to detect more minority drug resistance variants than Sanger. There was a sharp increase in minority drug resistance variants when the detection threshold was below 5%.

## 1. Introduction

The scale-up of antiretroviral therapy (ART) has reduced HIV-related deaths and prevented new HIV infections [1]. By the end of 2019, 25.4 million people globally had received ART, with an increase of 19 million when compared with 2009 [2]. However, while the number of patients with ART increases, so does the number of patients with treatment interruption. The rate of ART discontinuation ranges from 10% to 78% under different settings [3,4,5,6] and keeps rising with ART prolonged [7]. Patients with ART interruption have decreased CD4^+^ T cell counts [8], a higher risk of AIDS or death [9] and are a potential source of HIV transmission.

The emergence of HIV drug resistance (HIVDR) results from the low fidelity of HIV reverse transcriptase, the rapid replication of the virus and the selective pressure of antiretroviral drugs [10]. It will compromise the efficacy of ART, lead to virological failure and hamper the progress of HIV/AIDS treatment and prevention [11]. Under ART interruption, HIVDR variants may persist, or revert to wild-type strains or to a resistant revertant like T215rev [12,13]. In addition, new HIVDR variants may be selected by residual drugs with longer half-lives in combined antiretroviral regimens [14].

HIVDR assays are usually carried out using Sanger sequencing (SS), which can detect minority variants at a 15%–20% frequency in HIV viral populations (quasi species) within patients [15]. Next-generation sequencing (NGS) has been increasingly valued in recent years, having the ability to detect lower-frequency mutants and thus more HIVDR variants [16], with increased throughput and higher cost-efficiency [17]. The HIVDR mutation frequencies detected by NGS, but not SS, concentrate between 1.1% and 21.3% [18]. NGS could identify HIV drug-resistant variants at a frequency as low as 0.4% [19]. When patients with low-frequency HIVDR mutations receive ART again, the minority drug-resistant strains may return as predominant ones under the selective pressure of the drug [20]. In addition, multiple studies have shown that the presence of low-frequency HIVDR mutations is often related to treatment failure [21,22].

In this study, we conducted a cross-sectional survey among patients under ART interruption, and compared the mutation detection between NGS and SS using plasma samples from patients with ART interruption, to provide more information about detecting low-frequency drug resistance mutations and further assistance in implementing ultrasensitive HIVDR surveillance in routine assays and to guide the choice of treatment regimen.

## 2. Results

### 2.1. Characteristics of the Study Population

A total of 174 patients were included in the survey in 2016 (Table 1). Only 60 (34.5%) patients were followed up a median 10 months later, with 49 still in ART interruption and 11 re-initiating ART. Among the 174 patients, 88.5% (154/174) were aged 30 and above; 66.1% (115/174) were male; 65.5% (114/174) were illiterate or had a primary school education; 67.2% (117/174) were married or living with a partner; 61.5% (107/174) and 4.0% (7/174) of the patients were infected through heterosexual and homosexual contacting, respectively and 27.6% (48/174) were infected through injection drug using. The numbers of patients with CRF01_AE, CRF07_BC and CRF08_BC HIV-1 strains were 35, 61 and 58 (20.1%, 35.1% and 33.3%), respectively. Seventy (40.2%) patients had a CD4^+^ T cell count of <200 cells/mm^3^ at investigation. A total of 150 (86.2%) patients had received non-nucleotide reverse transcriptase inhibitor (NNRTI)-based first-line antiretroviral regimens (stavudine (d4T)/azidothymidine (AZT)/tenofovir (TDF) + lamivudine (3TC) + efavirenz (EFV)/nevirapine (NVP)) before ART interruption. The median duration of treatment before ART interruption was 16 (interquartile range (IQR): 7–26) months, while the median duration of ART interruption was 12 (IQR: 6–24) months at survey.

### 2.2. Detected HIV Drug Resistance Mutations

Drug resistance mutations (DRMs) were detected at 12 positions in the partial *pol* region by Sanger sequencing (Figure 1), including one in the protease (PR) region, an accessory protease inhibitor (PI)-related resistance mutation (Q58E), and eleven in the reverse transcriptase (RT) region, including one nucleotide reverse transcriptase inhibitor (NRTI)-related (M184V), and ten others were NNRTI-related. The most common drug resistance mutation was K103N (13.2%), followed by V179D (6.3%) and E138AGK (3.5%). At the 20% detection threshold, NGS detected all the mutations identified by SS except a V106M mutation in one patient, as there were mixtures in the first (G to R) and third nucleotides (A to R) in the codon at Sanger sequencing. In addition, NGS detected three more mutations in three patients: Y188C, E138A and K103N, at frequencies of 22.96%, 27.38% and 43.68%, respectively. There were no additional DRMs detected at the frequencies between 15% and 20%. Low-frequency DRMs (<15% frequency) were only detected by NGS, but not by SS. K65R was the most common low-frequency DRM with frequencies between 1% and 9%, concentrated at frequencies from 2% to 5%. It is interesting that this low-frequency K65R mutation is significantly unevenly distributed among subtypes; 5.7% (2/35) in CRF01_AE, when compared with 95.1% (58/61) in CRF07_BC and 93.1% (54/58) in CRF08_BC (*p* < 0.001). Other low-frequency NRTI-related mutations were K70QE, F77L, T215AI and K219QE. M46LI was the common low-frequency PI-related mutation, with frequencies ranging from 1% to 13%. Other PI-related mutations such as L10F, I47V, I50V, F53L, I54VT and N83D have the mutation frequency of about 2%.

### 2.3. HIV Drug Resistance Interpretations

Based on Sanger sequencing, 19.5% (34/174) of the patients were shown to have drug resistance variants. With NGS, the rate of resistance was the same; 20.7% (36/174) at the detection thresholds 20% and 15%. It climbed up to 21.3% (37/174), 24.1% (42/174), 45.4% (79/174) and 79.9% (139/174) at thresholds 10%, 5%, 2% and 1%, respectively. Compared with SS, NGS got significantly higher rates of drug resistance at the 1%, 2% and 5% thresholds (*p* < 0.05). For PI and NRTI, the prevalence of HIVDR was the same at 0.6%, identified by SS and NGS at the 15% detection threshold. However, NGS at the 1% and 2% thresholds identified more NRTI-related drug resistance variants (69.0% and 24.7%, respectively) than SS. Compared with NGS, a slightly lower percentage of HIVDR was found by SS for NNRTI-related drug resistance (19.0%, Table 2). For the efavirenz (EFV) or nevirapine (NVP) in first-line NNRTI, the difference between NGS and SS in the identification of drug resistance levels was relatively small. EFV- or NVP-related resistance rates were identified in 15.5% (27/174) and 16.1% (28/174) of patients by SS and NGS (>15% frequencies), respectively.

### 2.4. Relationship between CD4^+^ T Cell Count, Viral Load and HIVDR Mutation Frequency

The patients were divided into three groups across the mutation frequencies detected by NGS: 35 patients without HIVDR variants, 103 patients with mutation frequencies lower than 15% and 36 patients with mutation frequencies higher than 15%. Their median CD4^+^ T cell counts were 140 cell/mm^3^ (18–289), 265 cell/mm^3^ (IQR: 172–378) and 222 cell/mm^3^ (IQR: 82–302), respectively (*p* < 0.05). In addition, their median viral loads were 4.6 log_10_ copies/mL (4.0–4.9), 4.3 log_10_ copies/mL (IQR: 3.6–4.7) and 4.0 log_10_ copies/mL (IQR: 3.6–4.4), respectively (*p* < 0.05). In addition, there was a statistically significant difference in the viral loads between patients with high-frequency variants and patients without variants (*p* = 0.0198, Figure 2).

### 2.5. Changes of HIVDR Mutations at Follow-Up

Forty-nine patients were still with ART interruption and were available for follow-up after the median of ten months (IQR: 8–11). At baseline, mutations with a frequency of 20% and above were NRTI-related, such as M184VI (2.0%, 1/49), and NNRTI-related like K103N (14.3%, 7/49), E138AG (4.1%, 2/49), V179D (2.0%, 1/49) and P225H (2.0%, 1/49). Although these variants still existed at follow-up, the frequencies of the mutations M184VI, K103N and P225H decreased over time, and most of them remained at frequencies of more than 20%. However, the frequency of K103N in one patient (GX064) had dropped from 43.7% to 15.3%, and the mutation K103N in another patient (CQ046) with a frequency of 36.9% disappeared (Appendix A). Within a year, some minority DRMs at frequencies 1%–10% remained unchanged, including: PI-related D30N, M46LI, I54VT and N88D, NRTI-related K65R and NNRTI-related Y188CHL. Moreover, K65R was still the most common low-frequency mutation at the follow-up. However, some minority DRMs at frequencies of 1%–5% disappeared, including N83D with PI-related, K70E, T215A, and K219E with NRTI-related and K101E, Y181C, H221Y and K238T with NNRTI-related, while others emerged, such as NNRTI-related V106A in patient GX088 at a frequency of 8.7%, and L23I, I47V and I84V with PI-related, D67N and F77L with NRTI-related and L100V and F227L with NNRTI-related, which appeared at frequencies of 1%–5% (Figure 3).

## 3. Discussion

In this study, a cross-sectional survey was conducted among 174 patients with ART interruption in five areas more heavily affected by HIV/AIDS in China. The prevalence of HIVDR was 19.5% at Sanger sequencing, which was obviously higher than a pretreatment HIVDR survey conducted among ART naïve patients in 2017 [23]. A cross-sectional survey of patients with a median of 13.9 months on antiretroviral therapy in eight provinces in China showed that the prevalence of HIVDR in patients receiving ART was 4.3% [24]. Other studies have also suggested that the prevalence of drug resistance in patients receiving ART in some areas of China during the same period is less than 5% [25,26,27]. Compared with patients who have been receiving antiretroviral therapy, patients whose ART has been interrupted have a higher HIVDR prevalence. Furthermore, the rate of resistance to EFV and/or NVP (15.5%) was more than three times that (4.6%) in those patients. These findings were consistent with those in other countries [28], providing more evidence that patients with ART interruption should consider substituting EFV/NVP for PI or integrase inhibitors when re-initiating antiretroviral treatment.

It was unexpected to get higher rates of HIVDR at NGS than through Sanger sequencing. However, the increase in HIVDR was not statistically significant until the detection threshold was lowered to 5%. Moreover, DRMs detected in protease would have limited impacts on such PIs as Lopinavir/r (LPV/r) in second-line regimens. Four DRMs including D30N (1), M46I (2) and N88D (1) were detected in protease in four participants at a 5%–15% frequency, in addition to one Q58E at a frequency above 90%. The former two confer high or intermediate resistance to Nelfinavir, which is seldomly used at present. N88D and Q58E are PI accessary resistance mutations. There were slight increases in NRTI and NNRTI resistance at the threshold of 5%, when compared with the threshold of 20% or SS, which was consistent with other previous studies [29,30].

Despite the same primer sets for reverse transcription and PCR amplification, there were three NNRTI DRMs in three patients with frequencies between 23% and 44% detected by NGS, not SS. These discrepancies were more likely to be PCR resampling produced at the first several rounds of amplification, and less likely to be the artifacts during NGS sequencing, as NGS artifacts had a <1% frequency and were randomly distributed. Compared with that at the threshold of 5%, the rates of HIVDR were almost doubled when the threshold decreased to 2%, and tripled when down to 1%, as very-low-frequency DRMs were dramatically increased between 1% and 5%. The minority mutations detected by NGS may have resulted from apolipoprotein B mRNA editing catalytic polypeptide (APOBEC)-mediated G-to-A hypermutation such as E138K, M184I and G190E in RT or spontaneous mutation during viral replication, or from PCR error, which was introduced by error-prone reverse transcriptase or PCR enzymes [31]. The sharp increase in low-frequency DRMs agrees with the 5% detection threshold mentioned in other studies [32,33]. At present, the optimal threshold for the NGS detection of low-frequency mutation sites is still inconclusive. Studies have shown that a 5% threshold can provide a reproducible clinically relevant treatment result [34], suggested using a detection threshold of 5% to minimize technical artifacts [31,35]. Using a 5% threshold to report low-frequency variation allows NGS to achieve a moderately increased sensitivity to detect low-frequency DRMs, without compromising sequence accuracy [36,37].

The most common NNRTI DRM above a 5% frequency was K103N (14.4%) in RT, which is consistent with the ART history of EFV/NVP-based first-line therapy. In NRTI DRMs, K65R was the highest (1.2%) at a >5% frequency, which was similar to other research results [29,38]. In addition, this mutation was dramatically increased to 20.7% at 2%–5% frequency. We also found that the distribution of the K65R mutation was related to the HIV subtypes CRF07_BC and CRF08_BC. The K65R mutation causes high-level resistance to tenofovir and intermediate resistance to lamivudine, which are the backbone of first- and second-line regimens. The minority K65R mutations may become major mutations when re-initiating ART, giving rise to virological failure. The subtype-specific differences in the K65R mutation distribution are mainly due to differences in codon usage [39], as it is subtype C in this region in the CRF07_BC and CRF08_BC strains.

The disease progression in patients with HIV is usually monitored by measuring their CD4^+^ T cell counts. After the interruption of antiretroviral therapy, the CD4^+^ T cell count drops rapidly in the first few months, then declines slowly [40,41]. A previous study did not find a significant difference in the CD4^+^ T cell count between patients with and without HIVDR variants [42]. In this study, the patients were divided into three groups across mutation frequencies detected by NGS at a 15% threshold: patients without variants, patients with low-frequency variants and patients with high-frequency variants. Interestingly, the Kruskal–Wallis test found significant differences in the CD4^+^ T cell counts among the three groups. After the interruption of ART, the viral load level increased exponentially, reaching a peak of 10^6^ copies/mL. Then, the viral load usually drops to 10^3^–10^5^ copies/mL within a few weeks, which can remain relatively stable for many years [43]. In this study, the viral load of patients under ART interruption was above 10^3^ copies/mL, and there were statistical differences in the viral loads among the three groups.

Generally speaking, the drug resistance rate will decrease over time in patients after ART interruption. This trend is mainly due to the fact that, in the absence of drug selection pressure, wild-type strains with high viral fitness replace resistant strains to become the main strains in patients [44]. Compared with the baseline, the HIVDR mutations, such as N88D, K65R, M184VI, K103N, E138AG, V179D and P225H, still existed but the frequency gradually decreased, consistent with earlier studies [18,42]. Nevertheless, we found new low-frequency mutations such as I84V, F77L and V106A at follow-ups in this study. A study also showed that new mutations emerge in the absence of drug selection pressure [18]. The emergence of new mutations may be due to the reason that, without the drug selection pressure, virus strains carrying resistance-related mutations may have reduced replication fitness, so they cannot become dominant strains. The results of this study show that new mutations emerge in the form of low frequency, but whether they will continue to exist as the main drug-resistant strains over time is still unknown, and further research is needed.

There are limitations to this study. Firstly, the number of participants was limited. The findings could not be simply extrapolated to other patients with ART interruption. However, the fall or rise in DRMs would be similar to the other settings without ART. The minority DRMs detected by NGS were consistent with other pre-treatment surveys. Secondly, the participants in this study mainly used NNRTI-based first-line regimens before ART interruption. Thus, the findings were not able to reflect the situations with the interruption of ART on PI or integrase inhibitors. Thirdly, the rate of follow-up was low, as patients with ART interruption also dropped out of the HIV care program and were reluctant to keep the follow-up appointments.

In conclusion, patients with ART interruption had higher HIVDR, especially to EFV/NVP. NGS is able to detect more DRMs in such patients than SS, but largely at frequencies of less than 5%. Studies have shown that the drug resistance reports based on a 5% threshold can predict the failure of ART [21,45]. Due to the limitations of this study, the optimal NGS threshold for predicting the failure of antiviral therapy has not yet been determined. Further studies are needed to define the NGS detection threshold for providing a predicative of ART failure.

## 4. Materials and Methods

### 4.1. Study Design and Study Participants

We conducted a cross-sectional study among adult patients with ART interruption between 1 January and 31 December 2016, in four provinces and one municipality of China, including Guangxi, Henan, Sichuan, Yunnan and Chongqing, where in China the HIV epidemic was more concentrated, as described previously [46]. Inclusion criteria were: patients aged eighteen years or older, had discontinued ART for at least one month at survey, had received ART for at least three months before ART interruption and had agreed to provide written informed consent. These patients were followed up in 2017. Plasma samples were handled locally and transported in dry ice to the laboratory at the National Center for AIDS/STD Control and Prevention (NCAIDS), Chinese Center for Disease Control and Prevention (China CDC), where the HIV viral load was measured, and HIV drug resistance was genotyped on samples with a viral load ≥1000 copies/mL.

### 4.2. HIV RNA Extraction and Sample Amplification

Viral RNA was extracted using the QIAamp viral RNA mini kit (Qiagen, Hilton, Germany), following the manufacturer’s protocol. Extracted RNA was used for amplification of the PR region (4–99 amino acids) and the partial RT region (1–251 amino acids) of the HIV-1 *pol* gene region using an in-house method [47]. Briefly, the first-round PCR was performed in 25 µL volume reactions using the one-step RT-PCR kit (Promega, Madison, WI, USA) with the primers PRTM-F1 and RT-R1. The second-round PCR was performed in 50 µL volume reactions using the primers PRT-F2 and RT-R2.

### 4.3. Sanger Sequencing and Drug Resistance Analysis

The amplified products were sequenced using Sanger sequencer ABI 3730 (Applied Biosystems, Carlsbad, CA, USA). Obtained raw sequences were edited and assembled using Sequencher (version 4.10.1, Gene Codes Corporation, Ann Arbor, MI). The secondary peak threshold was set to 20%. The edited sequence was aligned with the HXB2 reference sequence through BioEdit (version 7.0.9, Informer Technologies Inc.). Sample contamination and other sequence quality controls were monitored using the WHO HIVDR QC tool (https://recall.bccfe.ca/who_qc/, accessed on 23 September 2019) and through constructing Neighbor joining phylogenetic trees using MEGA (version 6.06). HIV-1 subtypes were determined using reference sequences from HIV Databases (https://www.hiv.lanl.gov/content/index, accessed on 8 October 2019) through phylogenetic trees. The drug resistance mutations were identified and interpreted using the HIVdb version 8.9-1 (https://hivdb.stanford.edu/hivdb/, accessed on 10 October 2019).

### 4.4. Next-Generation Sequencing and Drug Resistance Analysis

Second-round amplicons were cleaned using KAPA PureBeads (Roche, Basel, Switzerland) and quantified using the Qubit dsDNA HS assay kit (Thermo Fisher Scientific, Carlsbad, CA, USA). Sequencing libraries were prepared using the 96-sample KAPA HyperPlus Kit (Roche, Basel, Switzerland). The samples were pooled in each run with a 40% Phix control library (v3, Illumina, San Diego, CA, USA) to increase the diversity of libraries and then sequenced on the Illumina Miseq system [48] using a v3 600-cycle reagent kit (Illumina, San Diego, CA, USA). The raw data were analyzed using the HyDRA Web tool (http://hydra.canada.ca/, accessed on 20 January 2020) according to *HyDRA Web User Guide* [49], producing lists and frequencies of drug resistance mutations, and then interpreted by using the HIVdb algorithm.

### 4.5. Statistical Analysis

DRMs were identified according to the 2019 International Antiviral Society (IAS) list. HIV drug resistance was defined as low-, intermediate- or high-levels of drug resistance according to the HIVdb algorithm 8.9-1. HIV drug resistance was determined based on NGS with detection thresholds of 20%, 15%, 10%, 5%, 2% and 1%, and was compared with that detected based on Sanger sequencing by the McNemar test, respectively. Quantitative data are presented as medians and were analyzed by the Kruskal–Wallis test. All data were analyzed using the Statistical Analysis System version 9.4 (SAS Institute Inc., Cary, NC, USA). Two-sided *p* values of <0.05 were considered statistically significant.

## Figures and Tables

**Figure 1 pathogens-10-00264-f001:**
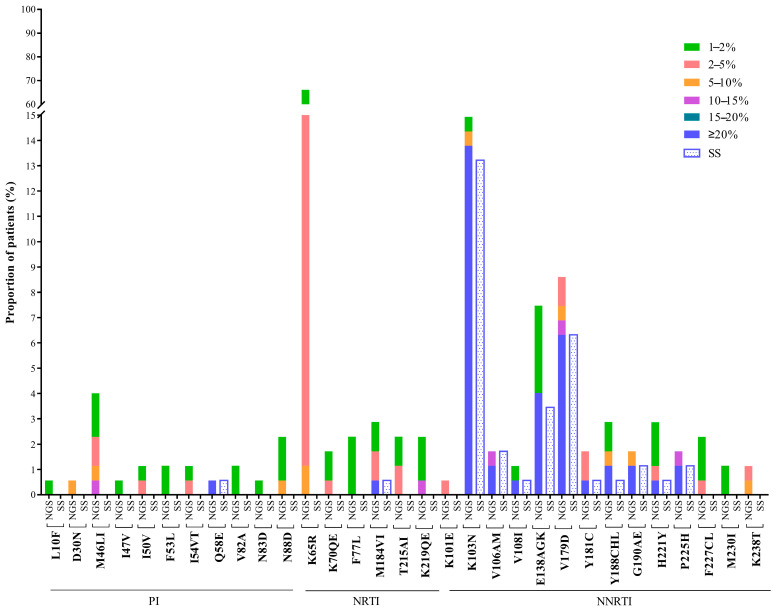
Frequency and pattern of HIV drug resistance (HIVDR) mutations detected by Sanger sequencing (SS) and next-generation sequencing (NGS) at different detection thresholds. Note: HIVDR, HIV drug resistance; SS, Sanger sequencing; NGS, Next-generation sequencing; PI, Protease inhibitor; NRTI, Nucleoside reverse-transcriptase inhibitor; NNRTI, non-nucleoside reverse-transcriptase inhibitor.

**Figure 2 pathogens-10-00264-f002:**
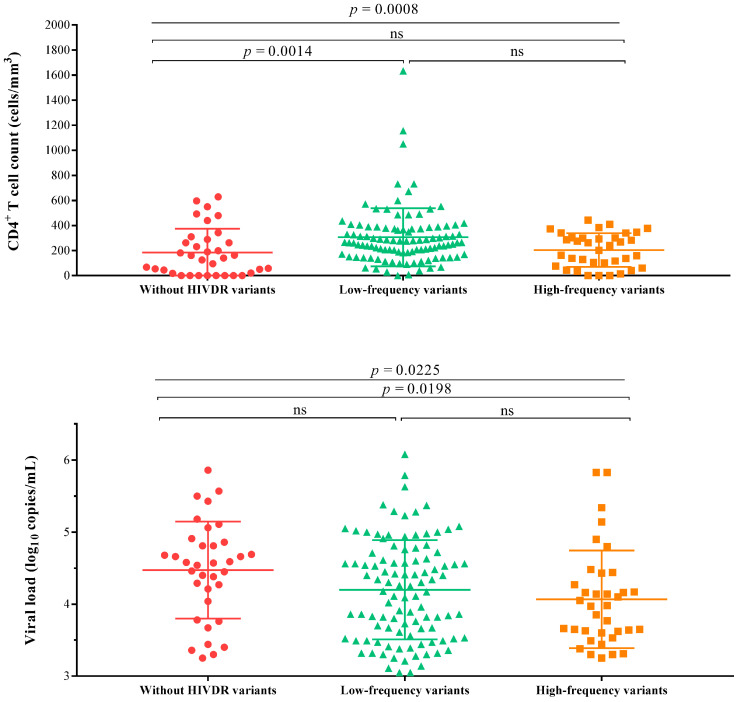
The relationship between the CD4^+^ T cell count, viral load and HIV drug resistance mutation frequency. Note: The patients were divided into three groups according to the mutation frequency detected by next-generation sequencing. High-frequency variants mean that their mutation frequencies were higher than 15% and low-frequency variants mean that their mutation frequencies were less than 15%. HIVDR, HIV drug resistance; ns, no significance.

**Figure 3 pathogens-10-00264-f003:**
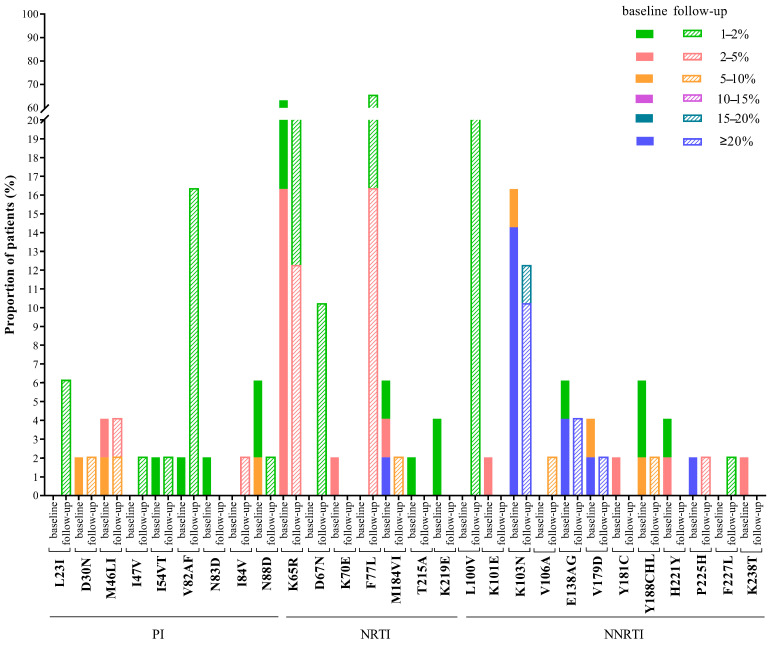
Changes of HIV drug resistance mutations during baseline and follow-up. Note: 49 of the 174 patients were followed up nearly a year later. Mutations and mutation frequency were detected by next-generation sequencing. PI, Protease inhibitor; NRTI, Nucleoside reverse-transcriptase inhibitor; NNRTI, non-nucleoside reverse-transcriptase inhibitor.

**Table 1 pathogens-10-00264-t001:** Characteristics of participants with antiretroviral therapy (ART) interruption.

Characteristics	Number (%)
Total	174
Age	
18–30	20 (11.5)
30–50	78 (44.8)
≥50	76 (43.7)
Gender	
Male	115 (66.1)
Female	59 (33.9)
Education	
Illiterate and primary school	114 (65.5)
Secondary school and above	60 (34.5)
Marital status	
Married or living with partner	117 (67.2)
Single	32 (18.4)
Other	25 (14.4)
Route of HIV infection	
Heterosexual	107 (61.5)
Homosexual	7 (4.0)
Injecting drug using	48 (27.6)
Other	12 (6.9)
Subtype	
CRF01_AE	35 (20.1)
CRF07_BC	61 (35.1)
CRF08_BC	58 (33.3)
B′	16 (9.2)
Other ^a^	4 (2.3)
CD4^+^ T cell count at the time of investigation(per uL)	
<200	70 (40.2)
200–350	62 (35.6)
≥350	42 (24.2)
Antiretroviral regimen before discontinuation	
d4T+3TC+EFV/NVP	1 (0.6)
AZT+3TC+EFV/NVP	66 (37.9)
TDF+3TC+EFV/NVP	83 (47.7)
AZT/TDF+3TC+LPV/r	24 (13.8)
Duration of treatment before ART interruption(median, (IQR), months)	16 (7–26)
Duration of ART interruption at survey (median, (IQR), months)	12 (6–24)

^a^ includes CRF 55_01B (2, 1.1%), CRF 62_BC (1, 0.6%), Unknown (1, 0.6%). HIV, Human immunodeficiency virus; ART, antiretroviral therapy; IQR, interquartile range; d4T, Stavudine; 3TC, Lamivudine; AZT, Azidothymidine; TDF, Tenofovir; EFV, Efavirenz; NVP, Nevirapine; LPV/r, Lopinavir/r.

**Table 2 pathogens-10-00264-t002:** Prevalence of HIV drug resistance detected by SS and NGS.

	SS	20% NGS ^a^		15% NGS ^a^		10% NGS ^a^		5% NGS ^a^		2% NGS ^a^		1% NGS ^a^	
SS or NGS at Various Thresholds	N (%)	N (%)	*p* Value	N (%)	*p* Value	N (%)	*p* Value	N (%)	*p* Value	N (%)	*p* Value	N (%)	*p* Value
Any classes	34 (19.5)	36 (20.7)	0.317	36 (20.7)	0.317	37 (21.3)	0.180	42 (24.1)	0.011	79 (45.4)	<0.001	139 (79.9)	<0.001
PI-related	1 (0.6)	1 (0.6)	1.000	1 (0.6)	1.000	2 (1.2)	0.317	5 (2.9)	0.046	9 (5.2)	0.005	23 (13.2)	<0.001
NRTI-related	1 (0.6)	1 (0.6)	1.000	1 (0.6)	1.000	1 (0.6)	1.000	3 (1.7)	0.157	43 (24.7)	<0.001	120 (69.0)	<0.001
NNRTI-related	33 (19.0)	35 (20.1)	0.317	35 (20.1)	0.317	35 (20.1)	0.317	36 (20.7)	0.180	43 (24.7)	0.004	51 (29.3)	<0.001
EFV/NVP	27 (15.5)	28 (16.1)	0.564	28 (16.1)	0.564	29 (16.7)	0.317	31 (17.8)	0.103	37 (21.3)	0.004	46 (26.4)	<0.001

SS, Sanger sequencing; ^a^ NGS, next-generation sequencing at 1%, 2%, 5%, 10%, 15%, or 20% detection thresholds; PI, Protease inhibitor; NRTI, Nucleoside reverse-transcriptase inhibitor; NNRTI, non-nucleoside reverse-transcriptase inhibitor.

## Data Availability

Data are contained within the article.

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
