# Peer review of "HIV Drug Resistance Mutations Detection by Next-Generation Sequencing during Antiretroviral Therapy Interruption in China"

_pathogens, 2021, doi:10.3390/pathogens10030264_

Round 1
Reviewer 1 Report
Authors reported findings from a cross-sectional study on the prevalence of HIV drug resistance variants/mutations detected by Sanger and NGS technologies among a cohort of HIV patients with >3 month ART interruption. Study found that the HIV DRMs detected by NGS were almost identical to Sanger at 20% threshold, and similar at the 10-15% but higher at threshold of ≤5%. Three comments are showed below:
- Study found 19.5% of HIVDR prevalence rate among patients who interrupted ART for median 12 months. As a cross sectional study, it would be very interested to know what was the prevalence rate of those HIV patients who didn't not interrupt ART during the same study time period. With this data, it would make the conclusion of "Patients with ART interruption had higher HIVDR" stronger and comparable.
- Authors observed the HIVDR variants at threshold of 5% or lower "come and go" among the patients with continuously interrupted ART. These variants at low abundance could be the test artifacts, or random mutations with or without advantage in viral fitness or clinical impacts. Because of that, it is debating that if the threshold at 5% should be used for reliable calls for HIVDR variants. Authors may cite or comment the consensus of accepted/recommended threshold for NGS HIVDR minor variant calls.
- In the Introduction, authors should indicate the AIDS/HIV epidemic data is global data.
- KAPA Hyper kit is not made by illumina in the USA. Please correct.
Author Response
Point 1: Study found 19.5% of HIVDR prevalence rate among patients who interrupted ART for median 12 months. As a cross sectional study, it would be very interested to know what was the prevalence rate of those HIV patients who didn't not interrupt ART during the same study time period. With this data, it would make the conclusion of "Patients with ART interruption had higher HIVDR" stronger and comparable.
Response 1: In some areas of China, the drug resistance rate of HIV patients who have not interrupted antiretroviral therapy during the same study period is basically less than 5%, which is in a low-level drug resistance state. Additions and modifications have been made in lines 186-192 of the manuscript.
Point 2: Authors observed the HIVDR variants at threshold of 5% or lower "come and go" among the patients with continuously interrupted ART. These variants at low abundance could be the test artifacts, or random mutations with or without advantage in viral fitness or clinical impacts. Because of that, it is debating that if the threshold at 5% should be used for reliable calls for HIVDR variants. Authors may cite or comment the consensus of accepted/recommended threshold for NGS HIVDR minor variant calls.
Response 2: Although the optimal threshold for the interpretation of low-frequency mutations has not yet been determined, most studies recommend a threshold of 5%. Additions and modifications have been made in lines 219-224 of the manuscript.
Point 3: In the Introduction, authors should indicate the AIDS/HIV epidemic data is global data.
Response 3: Additions and modifications have been made in line 50 of the manuscript.
Point 4: KAPA Hyper kit is not made by illumina in the USA. Please correct.
Response 4: Modifications have been made in line 316 of the manuscript.
Reviewer 2 Report
Comments:
The manuscript “HIV Drug Resistance Mutations Detection by Next-generation Sequencing During Antiretroviral Therapy Interruption” from Li et al. it is well written.
The paper analyzes the HIV drug resistance mutations through Sanger and next-generation sequencing in parallel, in patients with ART interruption in five regions in China in 2016, the rates of HIV drug resistance obtained through both methods were compared .
The study contributes for the knowledge of the incidence of HIV drug resistance in China and the mutations envolved.
A suggestion and a question:
- I think will be more informative if in the title you refer that the study was performed in China.
- Since NSS was able to detect more minority drug resistance variants than SS mostly when the detection threshold was below 5%, in your opinion what should be the NGS detection threshold for providing predicative of ART failure.
Author Response
Point 1: I think will be more informative if in the title you refer that the study was performed in China.
Response 1: The title has been revised to indicate that this research was conducted in China.
Point 2: Since NSS was able to detect more minority drug resistance variants than SS mostly when the detection threshold was below 5%, in your opinion what should be the NGS detection threshold for providing predicative of ART failure.
Response 2: The patients included in this study have less data after re-treatment, and the optimal threshold that can predict ART failure cannot be obtained. We will conduct further studies and analysis based on the drug resistance and virological data of patients before and after treatment in order to determine the optimal threshold. However, studies have shown that reports of low-frequency mutations based on the 5% threshold can predict virological failure. Additions and modifications have been made in lines 274-277 of the manuscript.